# A Cocktail of Lipid Nanoparticle-mRNA Vaccines Broaden Immune Responses against β-Coronaviruses in a Murine Model

**DOI:** 10.3390/v16030484

**Published:** 2024-03-21

**Authors:** Yi Zhang, Jialu Zhang, Dongmei Li, Qunying Mao, Xiuling Li, Zhenglun Liang, Qian He

**Affiliations:** 1Division of Hepatitis and Enterovirus Vaccines, Institute of Biological Products, National Institutes for Food and Drug Control, NHC Key Laboratory of Research on Quality and Standardization of Biotech Products, NMPA Key Laboratory for Quality Research and Evaluation of Biological Products, State Key Laboratory of Drug Regulatory Science, Beijing 102629, China; zhangyi199809@163.com (Y.Z.); zhang1999jialu@163.com (J.Z.); 2Shanghai Biological Products Research Institute Co., Ltd., State Key Laboratory of Novel Vaccines for Emerging Infectious Diseases, Shanghai 200052, China; ahal@163.com; 3National Engineering Laboratory for AIDS Vaccine, School of Life Sciences, Jilin University, Changchun 130012, China

**Keywords:** mRNA vaccine, coronavirus, broad-spectrum vaccine, cocktail immunization, sequential immunization

## Abstract

Severe acute respiratory syndrome (SARS)-coronavirus (CoV), Middle Eastern respiratory syndrome (MERS)-CoV, and SARS-CoV-2 have seriously threatened human life in the 21st century. Emerging and re-emerging β-coronaviruses after the coronavirus disease 2019 (COVID-19) epidemic remain possible highly pathogenic agents that can endanger human health. Thus, pan-β-coronavirus vaccine strategies to combat the upcoming dangers are urgently needed. In this study, four LNP-mRNA vaccines, named O, D, S, and M, targeting the spike protein of SARS-CoV-2 Omicron, Delta, SARS-CoV, and MERS-CoV, respectively, were synthesized and characterized for purity and integrity. All four LNP-mRNAs induced effective cellular and humoral immune responses against the corresponding spike protein antigens in mice. Furthermore, LNP-mRNA S and D induced neutralizing antibodies against SARS-CoV and SARS-CoV-2, which failed to cross-react with MERS-CoV. Subsequent evaluation of sequential and cocktail immunizations with LNP-mRNA O, D, S, and M effectively elicited broad immunity against SARS-CoV-2 variants, SARS-CoV, and MERS-CoV. A direct comparison of the sequential with cocktail regimens indicated that the cocktail vaccination strategy induced more potent neutralizing antibodies and T-cell responses against heterotypic viruses as well as broader antibody activity against pan-β-coronaviruses. Overall, these results present a potential pan-β-coronavirus vaccine strategy for improved preparedness prior to future coronavirus threats.

## 1. Introduction

The coronaviruses have caused several pandemics in the 21st century and would continue to be a threat to human health in future. Coronavirus is divided into four genera, α, β, γ, and δ, and β-coronaviruses are composed of four lineages, including A, B, C, and D. The B lineage virus is also called sarbecovirus, including SARS-CoV, SARS-CoV-2, and its variants, and SARS-related coronaviruses (SARSr-CoVs) from bats and other animals [1]. The C lineage viruses, also known as merbecovirus, include MERS-CoV and MERS-related coronaviruses (MERSr-CoVs) originating from bats [2]. SARS-CoV, MERS-CoV, and SARS-CoV-2 of the β genus can be transmitted directly or indirectly from person to person, animal to person, or animal to animal and cause serious diseases in humans [3]. In the 21st century, SARS-CoV, MERS-CoV, and SARS-CoV-2 have emerged successively, and caused pandemics resulting in serious public health events [4,5]. SARS-CoV-2 rapidly spread from the end of 2019 and developed into a devastating pandemic, causing millions of deaths in recent years. Transmissible human coronaviruses continue to emerge and re-emerge, mutating as they spread, just as multiple variants circulated iteratively during the COVID-19 pandemic. A broad vaccine strategy to cope with emerging and re-emerging high-pathogenic coronaviruses is thus urgently needed [6].

Coronaviruses bind host cell receptors via spike proteins to invade them. Thus, spike protein is the most important target for developing effective vaccines against highly pathogenic coronaviruses [7,8]. Vaccines based on coronavirus spike proteins can effectively induce neutralizing antibodies against coronaviruses. However, the amino acid sequences of spike proteins differ to varying degrees among heterotypic coronaviruses (~76% identity between SARS-CoV-2 and SARS-CoV, ~35% identity between SARS-CoV-2 and MERS-CoV). Further, the host receptors recognized by sarbecoviruses and merbecoviruses are divergent (hACE2 for SARS-CoV, SARS-CoV-2, and DPP4 for MERS-CoV). Thus, the prototype vaccine candidates against SARS-CoV, SARS-CoV-2, and MERS-CoV cannot provide full protection against all three major pathogenic β-coronaviruses.

Two dominant strategies have been used to develop a broad-spectrum vaccine against coronaviruses: (i) simultaneous immunization with heterotypic antigenic components and (ii) the use of a conserved or shared antigenic sequence design. For the pan-SARS-CoV-2 vaccine, the prototype vaccine maintained activity against pre-Omicron variants, albeit with varying degrees of reduction. However, as Omicron and its sublineages vary, the Omicron component should be incorporated into the prototype vaccine to achieve a broad immune response against pan-SARS-CoV-2 [9,10]. Cross-antibodies and T-cell responses also exist between SARS-CoV-2 and other sarbecoviruses, such as SARS-CoV [11,12,13]. Thus, conserved antigenic epitopes have been introduced into pan-sarbeco vaccines in some studies [14,15]. However, the RBD amino acid sequences of MERS-CoV and SARS-CoV/SARS-CoV-2 are significantly different; for this reason, it is difficult to design a shared antigenic sequence to induce sufficient cross-protection against both pathogenic merbecoviruses and sarbecoviruses [16,17,18,19].

Therefore, developing an effective broad-spectrum vaccine to cope with all future SARS-like or MERS-like pandemics remains challenging. In this study, four strain-specific LNP-mRNA vaccines against β-coronaviruses were prepared and evaluated both in vitro and in vivo; further, sequential and cocktail immunization regimens were adopted to improve the broad-spectrum immunity against pathogenic pan-β-coronaviruses.

## 2. Materials and Methods

### 2.1. Cells and Animals

The human lung cancer cell line A549 (CRM-CCL-185) used in this study was purchased from ATCC and stocked in our laboratory. A549 cells were maintained in DMEM supplemented with 10% fetal bovine serum (FBS) and 1% Pen-Strep, and were cultured at 37 °C and 5% CO_2_. SPF-grade BALB/c mice (6–8-week-old, female, 18–22 g) were provided by and housed at the Laboratory Animal Resource Center, National Institute for Food and Drug Control and were used as experimental animals.

### 2.2. Preparation and Characterization of LNP-mRNA Vaccines

The spike sequences of three highly pathogenic coronaviruses, SARS-CoV (NC_004718), MERS-CoV (NC_019843), and SARS-CoV-2 (Pfize BNT162b) (OR134577) were obtained from the NCBI database. The BA.4&5 and Delta mutation sites were published by Cov-Lineages.org (accessed on 18 March 2024) [20], and are shown in Appendix A. All the coding sequences were optimized to adapt the nuclear codon pattern of eukaryote and cloned in the pUC57 vector, which contained a poly (A) tail (110 repeated A (adenine) added after 3’UTR), 5’UTR, and 3’UTR. The linearized plasmid template that encodes spike sequences of O (SARS-CoV-2 Omicron), D (SARS-CoV-2 Delta), S (SARS-CoV), or M (MERS-CoV) was transcribed into mRNAs in vitro using T7 RNA polymerase (YEASEN, Shanghai, China) and the Cap1 system. The uridine triphosphate (UTP) required for transcription was replaced with pseudouridine (Vaccinations, Cangzhou, China). The products were purified using column chromatography and stored at −80 °C. RNA concentrations were detected using NanoDrop (Thermo Fisher Scientific, Waltham, MA, USA), and the integrity and purity of the mRNA stock solution were evaluated using Qsep 100 (Bioptic, Changzhou, China). Subsequently, the mRNAs were encapsulated in liposomes [21]. We then determined the average particle size of the LNP-mRNAs [22] and evaluated their liposomal encapsulation rate by Quant-iT™ RiboGreen^®^ RNA Reagent and Kit (Thermo Fisher Scientific, Waltham, MA, USA. cat #R32700). The dsRNA residues were measured using dsRNA residue detection kit (Vazyme, Nanjing, China. cat # DD3508-01). All the LNP-mRNAs were stored at −20 °C.

### 2.3. mRNA Transfection and Western Blot Assay

A549 cells were seeded (1 × 10^6^ cells/well) in 6-well plates until 80% confluency; next, 3 μg of each of the four liposome-encapsulated vaccines was transfected into the 6-well plates. After 48 h of transfection, A549 cells were lysed by 10% SDS-polyacrylamide gels (Genescript, Nanjing, China) and transferred onto nitrocellulose membranes. The nitrocellulose membranes were blocked in 1×PBST containing 5% skim milk powder for approximately 2 h. The nitrocellulose membranes were incubated with mouse anti-SARS-CoV-2 (2019-nCoV) spike, mouse anti-SARS-CoV spike, and rabbit anti-MERS-CoV spike antibodies (Sino Biological, Beijing, China) for 2 h, and then incubated with HRP-labeled goat anti-mouse antibody (ZSGB-Bio, Beijing, China) or HRP-labeled goat anti-rabbit antibody (Cell Signaling Technology, Danvers, MA, USA) for 1 h after washing. An Amersham Imager 680 (Cytiva, Tokyo, Japan) was used for image capture.

### 2.4. ELISA for Spike-Specific IgG

Spike-specific IgG antibodies were detected using ELISA. The spike proteins of different pathogenic β-coronaviruses were individually coated at 1 μg/mL on 96-well EIA/RIA plates overnight at 4 °C. This was followed by blocking with PBS containing 10% FBS and 5% sucrose for 2 h at 37 °C the next day. The samples were diluted appropriately, added to the antigen-coated wells and incubated for 1 h at 37 °C. HRP-labeled goat anti-mouse IgG (ZSGB-BIO, Beijing, China. cat # ZB2305) was diluted 10,000-fold and added. Finally, color-developing solution (Wantai BioPharm, Beijing, China. cat # N20200722) was added and absorbance was recorded at dual wavelengths of 450 and 630 nm.

### 2.5. Serum Neutralization Assay

Serum-neutralizing antibodies were detected using authentic viruses and pseudoviruses. Authentic viruses included the SARS-CoV-2 variants BA.5, XBB.1.5, XBB.1.16, Delta, and WT (Wuhan Institute of Biological Products, Wuhan, China). The pseudoviruses included SARS-CoV-2 variants BA.4&5, XBB.1.16, Delta, SARS-CoV, and MERS-CoV (Yunling Biotec, Beijing, China.).

### 2.6. IFN-γ ELISPOT Assay

Mouse splenocytes were plated at 2 × 10^5^ cells/well and stimulated with spike protein peptide pools (5 μg/mL). After incubating at 37 °C in 5% CO_2_ for 20 h, the supernatant was removed. IFN-γ positive cells were detected by IFN-γ ELISPOT kit (BD, cat # 551083) and counted by an ELISPOT reader (CTL-ImmunoSpot^®^ S6, Cleveland, OH, USA). The final result was determined after background subtraction.

### 2.7. Statistical Analyses

All statistical analyses were conducted using GraphPad Prism 8 (GraphPad Software, Inc. Boston, MA, USA). Antibody titers were log-transformed and presented as the geometric mean with geometric standard derivation (geometric SD). Tukey’s multiple comparisons test was used for the comparison of antibody responses in different groups. ELISPOT data were presented as the mean with SD, and compared using Dunn’s multiple comparison test. *p*-value < 0.05 was considered statistically significant.

## 3. Results

### 3.1. Design and Characterization of Four Monovalent mRNA Vaccines against High-Pathogenic β-CoVs

Spike protein on the surface of coronaviruses is an important vaccine target, as it mediates the entry of coronaviruses into the organism for replication and translation [23,24]. In this study, the full-length spike mRNA of the SARS-CoV-2 (hereafter labeled SARS-2) Omicron BA.4&5 variant (BA.4&5), Delta variant (Delta), SARS-CoV (SARS), and MERS-CoV (MERS) was synthesized with a 5’UTR, 3’UTR, and poly A tail and encapsulated in LNPs to produce four LNP-mRNA vaccines denoted as O, D, S, and M, respectively (Figure 1A). Sequence alignment of the four spike sequences showed 97.3% identity between O and D, 75% identity between SARS and SARS-2, and less than 36% identity between SARS/SARS-2 and MERS (Figure 1B). The purity and integrity of the four vaccines were measured using capillary gel electrophoresis. All mRNAs showed a single sharp peak, with a purity of approximately 83% and the dsRNA residues were in the range of 1.29–1.58 μg/mg (Figure 1C). The target antigens were all correctly expressed in A549 cells (Figure 1D).

### 3.2. Target Antigen-Specific and Cross-Reactive Antibody Responses of the Four Monovalent mRNA Vaccines

To evaluate the in vivo immunogenicity of the four LNP-mRNAs, BALB/c mice were immunized with the prepared vaccines and their antibody responses were measured (Figure 2A). During the immune procedure, all the mice grew well with no decrease in body weight. The seropositivity rate after the first dose was 100% for all four LNP-mRNAs. Further, pseudovirus-neutralizing antibodies (NtAbs) against the corresponding virus were also stimulated in groups D, S, and M, but at low levels; and no NtAb activity was detected in group O. Regarding cross-reactivity, only LNP-mRNA D induced an NtAb response against the SARS spike. To further boost the antibody response in each group, we administered two booster immunizations. The IgG and pseudovirus NtAb titers gradually increased after boosting. After the second booster immunization, the antibody responses of the O, D, S, and M groups all increased with GMTs of 33,336, 30,824, 133,436, and 72,587 for the pseudovirus NtAb, respectively. On day 49, both LNP-mRNA O and LNP-mRNA D induced cross-binding antibodies against SARS and MERS spike antigens. LNP-mRNA D triggered higher cross-neutralizing activity against SARS than that of LNP-mRNA O (*p* < 0.0001). Although robust neutralizing activity was induced by LNP-mRNA D against SARS, NtAbs against SARS-2 were not induced by LNP-mRNA S (Figure 2B,C). Further, although cross-binding antibody titers against MERS spikes were induced by O, D, and S LNP-mRNAs, which gradually increased after booster immunization, the cross-binding ability of LNP-mRNA M against BA.4&5, Delta, and SARS was not effectively boosted. Moreover, the sera in groups O, D, and S showed extremely low neutralizing activity against the MERS pseudovirus, even after the second booster. This demonstrated poor cross-reactivity between the MERS and sarbecovirus spike antigens.

Furthermore, the neutralizing ability of the sera in groups O, D, S, and M was tested against several authentic viruses, including SARS-2 BA.5, XBB.1.5, XBB.1.16, and Delta, to evaluate their broad-spectrum protection against SARS-2 variants mediated by LNP-mRNA O, D, S, and M (Figure 2D). The results showed that LNP-mRNA O induced the highest NtAb responses against BA.5, XBB.1.5, and XBB.1.16, with GMTs of 10,033, 516, and 1192, which were 14.2- (*p* < 0.0001), 16.1- (*p* < 0.0001), and 8.7-fold (*p* < 0.0001) higher than those induced by LNP-mRNA D, respectively. Meanwhile, LNP-mRNA D induced a higher NtAb response against Delta, with a GMT of 6506, which was 5.2-fold (*p* = 0.0014) higher than that induced by LNP-mRNA O. However, the NtAb responses against SARS-2 viruses induced by LNP-mRNA D and M were significantly lower than those induced by LNP-mRNA O and D, showing NtAb GMTs ranging between 13 and 73.

### 3.3. The Target Antigen-Specific and Cross-Reactive T-Cell Responses of the Four Monovalent mRNA Vaccines

For spike-specific T-cell responses, splenocytes were harvested after the second booster and stimulated using peptide pools spanning the full-length spike of BA.1, Delta, SARS, and MERS (Figure 3A). The number of spike-specific IFN-γ-secreting T cells was then detected using an ELISPOT assay. Our results showed that all four mRNA vaccines—O, D, S, and M—induced specific T-cell responses against their respective spike peptide pools, with 409 (BA.1), 178 (Delta), 487 (SARS), and 639 (MERS) average spot forming units (SFUs). For cross-reactivity, 492 SFUs/10^6^ splenocytes were elicited using LNP-mRNA D against the BA.1 spike, and 145 SFUs/10^6^ splenocytes were elicited by LNP-mRNA O against Delta spike, comparable to that induced by their corresponding spike LNP-mRNA. This demonstrates that T cells activated by the Omicron spike and Delta spike are remarkably conserved. Furthermore, LNP-mRNA S and LNP-mRNA M also induced T cells recognizing the BA.1 spike and Delta spike. However, although O, D, and S could induce a certain number of specific T-cell responses against the MERS spike, the levels were significantly lower than those in group M, due to the poor amino acid sequence homology between the BA.1/Delta/SARS spike and MERS spike. Further, the supernatants of stimulated splenocytes were analyzed for Th1/Th2 cytokines (Figure 3B). The results showed that the four LNP-mRNAs mainly induced IFN-γ and IL-2 cytokines, indicating that the T-cell responses in the four groups were all Th1-skewed. In summary, LNP-mRNA O, D, S, and M effectively induced Th1-biased T-cell immune responses; further, the T-cell responses induced by SARS-2 spike LNP-mRNA were significantly reduced against the SARS and MERS spike proteins.

### 3.4. Sequential and Cocktail Immunization to Achieve a Broad Antibody Response against SARS-CoV-2, SARS-CoV, and MERS-CoV

To induce a broad-spectrum immune response against highly pathogenic β-CoVs, we attempted a sequential (S-M-OD) and cocktail (SMOD) immunization strategy with the O, D, S, and M vaccines (Appendix A). After priming with LNP-mRNA S, only NtAbs against the SARS pseudovirus were detected, with a GMT of 1043. After secondary immunization with LNP-mRNA M, serum NtAbs were significantly elevated by 67.5-fold (*p* < 0.0001) against MERS pseudovirus and by 19.6-fold (*p* < 0.0001) against SARS pseudovirus; NtAb responses against BA.4&5 and Delta pseudoviruses were also slightly elevated, but the differences were not significant (*p* = 0.9769 and *p* = 0.9852, respectively). After the third immunization with O and D, serum NtAb responses against BA.4&5, Delta, and SARS were all significantly elevated showing 14.8-fold (*p* = 0.0012), 2.94-fold (*p* < 0.0001), and 3.5-fold (*p* = 0.0002) higher levels than those after the second immunization, respectively (Figure 4B). The binding antibody results were similar to those of the pseudovirus NtAbs (Figure 4A). Therefore, immune protection could be enhanced and broadened using the S-M-OD sequential immunization strategy. Further, the dynamic immune responses induced by S-M-OD further confirmed the cross-immunity between SARS-2 and SARS, but poor cross-immunity between SARS-2/SARS and MERS.

Next, we compared the antibody responses induced by OD, S-M-OD, and SMOD. After the third immunization, we found high binding antibody titers against BA.4&5, Delta, SARS, and MERS spikes in the OD, S-M-OD, SMOD(Hd), and SMOD(Ld) groups. Compared to the OD group, S-M-OD (*p* = 0.9856), SMOD(Hd) (*p* = 0.0320), and SMOD(Ld) (*p* = 0.0353) elicited higher levels of binding IgG titers against MERS spike; among these, SMOD(Hd) (*p* = 0.6478) and SMOD(Ld) (*p* = 0.6705) elicited titers higher than those of the S-M-OD group. The sequential and cocktail immunization strategies did not further enhance binding antibody responses against BA.4&5, Delta, and SARS, and were even reduced (Figure 4A). Furthermore, we evaluated the serum-neutralizing activity in each group. The results showed that after the third immunization, the SMOD regimen did not further enhance, but maintained a considerably high level of pseudovirus NtAbs against Delta and BA.4&5 than that with the OD regimen. The NtAb GMTs against the SARS pseudovirus in the OD, S-M-OD, SMOD(Hd), and SMOD(Ld) groups were 23,664, 30,913, 123,148, and 99,395, respectively, and those against MERS pseudovirus were 207, 900, 15,522, and 32,002, respectively. Thus, compared to those with OD, immunization with S-M-OD, SMOD(Hd), or SMOD(Ld) effectively boosted NtAb responses against SARS and MERS. SMOD(Hd) and SMOD(Ld) both induced higher pseudoviral NtAb titers against BA.4&5, XBB.1.16, Delta, SARS, and MERS than those obtained with the S-M-OD group (Figure 4B).

We also measured the NtAbs against several authentic variants of SARS-2 (Figure 4C). Similar to the results of NtAbs against pseudoviruses, the OD group induced broad-spectrum NtAbs against XBB.1.15, XBB.1.16, and Delta, with NtAb GMTs of 163, 801, and 5671, respectively. The NtAb GMTs against BA.5, XBB.1.16, and Delta in the SMOD(Hd) group were 9.51-fold (*p* = 0.0002), 3.4-fold (*p* = 0.0324), and 6.9-fold (*p* < 0.0001) higher, respectively, than those in the S-M-OD group. However, there were no significant differences observed for XBB.1.5 between the two groups (*p* = 0.1380).

Therefore, when the number of immunizations was the same, cocktail immunization induced more powerful NtAb responses against SARS-2, SARS, and MERS than sequential immunization, even though the total dose of LNP mRNA was lower in the cocktail regimen.

### 3.5. Cocktail Immunization Elicited a Higher and Broader Th1 Cell Response

For the spike-specific T-cell responses, no significant difference existed in the quantity of BA.1- and Delta-specific T cells between the OD and SMOD groups (*p* > 0.9999). SARS- and MERS-specific T cells induced in the SMOD(Hd) group were higher than those in the OD group, with average SFUs/10^6^ 17-fold (*p* = 0.0193) and 106-fold (*p* = 0.0002) higher than those in the OD group, respectively. The SMOD(Hd) regimen induced higher levels of T-cell responses compared with those in the SMOD(Ld) and S-M-OD regimens. No significant difference was observed in the T-cell immune response induced by the SMOD(Ld) and S-M-OD regimens (Figure 5).

### 3.6. Cocktail Immunization Induced Robust and Broadly Reactive Antibodies against Pan β-CoVs

Zoonotic coronaviruses have continuously evolved to infect humans. In addition to SARS-2, SARS, and MERS, several other coronaviruses can also infect humans, including HCoV-229, HCoV-NL63, HCoV-HKU1, and HCoV-OC43 [4,25]. A phylogenetic tree was constructed to show the evolutionary relationships among the seven human coronaviruses (Figure 6A). We measured the IgG titers against these coronaviruses induced by the different immunization regimens described above. The results showed that (Figure 6B) OD, S-M-OD, and SMOD all induced high levels of binding antibody titers against β-CoVs HCoV-HKU1 and HCoV-OC43. Among all groups, the SMOD(Hd) regimen induced the highest binding IgG titers against HCoV-HKU1 and HCoV-OC43. Compared to that with S-M-OD, SMOD(Ld) induced a higher IgG GMT against HCoV-HKU1 and HCoV-OC43, which were elevated 2.3-fold (*p* > 0.1) and 1.7-fold (*p* > 0.1), respectively. As HCoV-229E and HCoV-NL63 belong to α-CoVs, all four groups showed low binding antibody titers against HCoV-229E and HCoV-NL63. These results demonstrated that the sequential and cocktail immunization with O, D, S, and M might be effective regimens to mediate broad protection against pan-β-CoV infection. Overall, cocktail immunization was superior to sequential immunization.

## 4. Discussion

Strain-specific vaccines against three major coronaviruses (MERS-CoV/SARS-CoV/SARS-CoV-2) have been developed based on various technological platforms, but none have provided broad protection against all three [26,27,28,29]. Here, we generated mRNAs encoding the spike proteins of Omicron, Delta, SARS, and MERS and encapsulated them in LNPs. These four LNP-mRNAs were characterized, evaluated for immunogenicity and cross-immunity, and examined for their ability to generate a broad-spectrum immune response against all pathogenic coronaviruses using sequential or simultaneous immunization strategies.

To provide clues for orchestrating broad-spectrum vaccine immunization strategies, we first analyzed cross-immune activity in mice. Mice were immunized with LNP-mRNAs encoding spike from Omicron, Delta, SARS, and MERS. The results showed that immunization with Omicron spike LNP-mRNA or Delta spike LNP-mRNA alone was difficult to induce a high NtAb response against both Omicron and Delta. This is consistent with the results of current reports that pre-Omicron variants, including WT, Alpha, Beta, Delta, possess a certain degree of cross-neutralizing activity to each other [30], and real-world data also showed that the first-generation SARS-2 vaccines targeting ancestral virus were still able to provide effective protection against the Alpha and Beta variants, whereas the Omicron variant gained more than 30 mutation sites and remains a poor cross-immune response to the pre-Omicron viruses [31]. Thus, both the pre-Omicron and Omicron components needed to be incorporated to generate a pan-SARS-CoV-2 vaccine. In fact, combined Omicron and Delta spike LNP-mRNA immunization (group OD) in this study did induce a broad spectrum of potent neutralizing antibodies and T-cell responses against all tested SARS-CoV-2 viruses.

In this study, Delta spike LNP-mRNA induced a certain level of NtAb against SARS. Similarly, recent studies have shown that SARS-2 spike or RBD protein can induce broad-spectrum antibody against sarbecovirus [32,33,34,35,36]. MERS spike, belonging to lineage C of β-coronaviruses, also elicited a certain degree of NtAb responses against SARS. In contrast, the SARS spike LNP-mRNA was ineffective at inducing NtAbs against Delta, Omicron, and MERS, suggesting that SARS spike has poor cross-immune activity against other β-coronaviruses. Further, LNP-mRNA encoding SARS-2 variants or SARS spike did not induce NtAb responses against MERS. This is attributed to (i) large differences in the MERS and SARS/SARS-2 spike sequences (sequence identity less than 35%), and (ii) different host receptors used for infection, with DPP4 for MERS and hACE2 for both SARS and SARS-2. Thus, few neutralizing epitopes are shared by SARS/SARS-2 and MERS spikes, hindering the development of a pan-β-CoV vaccine based on conserved neutralizing epitopes or shared immunogen sequences.

Currently, two strategies dominate pan-β-CoV vaccine development, including multivalent and chimeric vaccines against sarbecoviruses and merbecoviruses to achieve immunity against all pathogenic β-coronaviruses [37,38]. Using previously developed monovalent vaccines targeting various β-coronaviruses is a relatively simple approach for inducing broad-spectrum immune responses via immunization procedure orchestration. Studies on broad-spectrum HIV vaccine and influenza vaccines have confirmed the effectiveness of sequential and simultaneous immunization strategies [39,40]. Specifically, sequential immunization with immunogens from virus variants can achieve broad-spectrum immunity through repeatedly boosting and prioritizing immunity against conserved antigen epitopes; simultaneous immunization with different immunogens in a prime-boost regimen facilitates this by gradually strengthening the immune response against all dominant antigenic epitopes. In this study, S-M-OD and SMOD regimens both induced broad neutralizing antibodies against SARS, SARS-2, and MERS; among these, NtAb titers in the SMOD group are higher. Current studies on the correlates of protection (CoP) against coronaviruses state that a long-lasting and neutralizing antibody is an essential immune effector [41]. “High binding and low neutralizing” might mediate antibody-dependent enhancement. Thus, we agree that the SMOD regimen has more potential for immune protection against SARS, SARS-2, and MERS. However, due to limitations in experimental materials and methods, we were unable to detect the NtAb titers against other human coronaviruses. It should be addressed that further exploration is also needed to determine the protective efficacy in animal models.

Since the SARS spike did not induce desirable NtAbs against SARS-2 and MERS, we speculated that the SARS spike was not an effective booster for the SARS-2 or MERS spike. Thus, the sequential immunization regimen was orchestrated as S-M-OD—SARS spike for priming, and MERS and SARS-2 for boosting—to strengthen the conservative immune response. The results showed that the S-M-OD regimen progressively strengthened serum-neutralizing antibody responses against SARS, MERS, and SARS-2. However, whether NtAbs against conserved epitopes were induced in this study requires further investigation. Burnett et al. showed that immunizing humanized mice with diverse sarbecoviral RBDs could elicit rare neutralizing cross-reactive antibodies targeting the conserved class 4 epitope [42]. Therefore, we speculated that most NtAbs elicited in the S-M-OD group had no cross-neutralizing activity, but were a mix of various NtAbs against SARS, SARS-2, and MERS.

Overall, this study presents four LNP-mRNA vaccines targeting SARS, SARS-2, and MERS, which induce higher NtAb responses against SARS, SARS-2, and MERS upon simultaneous immunization than upon sequential immunization. However, all these results were generated in a mouse model, and need to be validated in primates for further application. Additional exploration is also required to probe the protective effects of the aforementioned broad-spectrum vaccine strategies against other pathogenic β-coronaviruses.

## Figures and Tables

**Figure 1 viruses-16-00484-f001:**
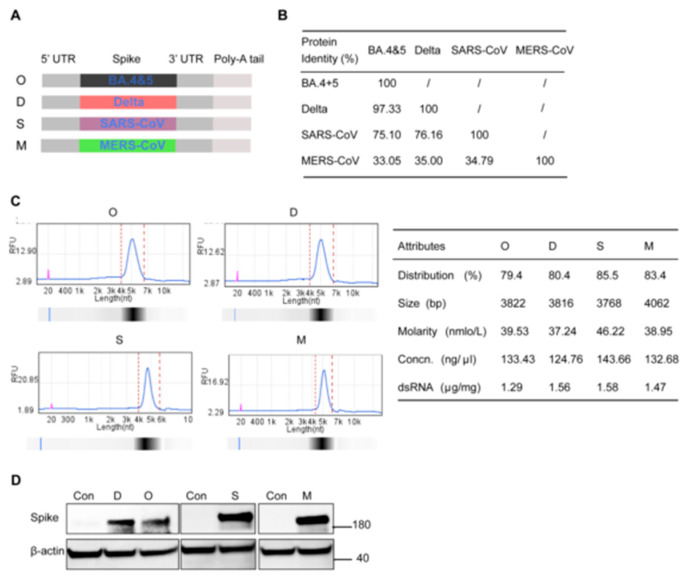
Design and characterization of four monovalent mRNA vaccines against high-pathogenic β-CoVs. (**A**) Skeleton diagram for the designation of the four LNP-mRNA vaccines. All four mRNAs contained the same 5’UTR, 3’UTR, poly A, and spike ORF. The ORF contains the spike proteins of BA.4&5, Delta, SARS, and MERS, respectively. (**B**) Amino acid sequence identity among SARS-CoV-2 BA.4&5, SARS-CoV-2 Delta, SARS-CoV, and MERS-CoV spike proteins. (**C**) mRNA vaccine (O, D, S, M) solutions were tested using Bioptic 100 and dsRNA residues were detected using a dsRNA residue detection kit. (**D**) The expression of the four mRNA vaccines in A549 cells was examined by Western blot analysis.

**Figure 2 viruses-16-00484-f002:**
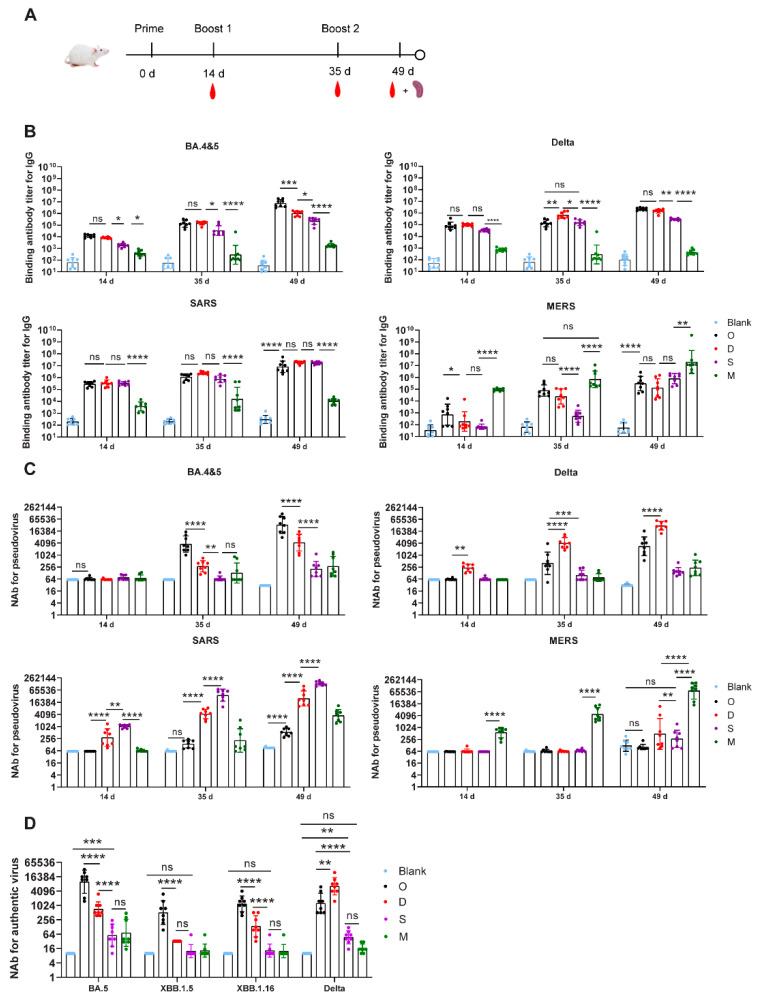
Antibody response induced by the four monovalent mRNA vaccines. (**A**). Schematics of vaccine schedule for the four monovalent mRNA vaccines. A total of three immunizations were performed and each mouse was injected intramuscularly in hind legs with 6μg LNP-mRNA/dose. The serum collection was performed on day 14, 35, and 49 after the initial immunization; splenocytes were collected on day 49. Serum spike-specific IgG titers (**B**) and neutralizing antibody (NtAb) titers against pseudovirus (**C**) were measured on days 14, 35, and 49. NtAb titers against live viruses were measured against the SARS-2 BA.4&5, XBB.1.15, XBB.1.16, and Delta strains on day 49 (**D**). *n* = 8 per group. Tukey’s multiple comparisons test was used to compare differences for (**B**–**D**), respectively. Bars represent the geometric mean with geometric SD, ns, *p* > 0.05; *, *p* < 0.05; **, *p* < 0.01; ***, *p* < 0.001; ****, *p* < 0.0001. IgG, immunoglobulin G.

**Figure 3 viruses-16-00484-f003:**
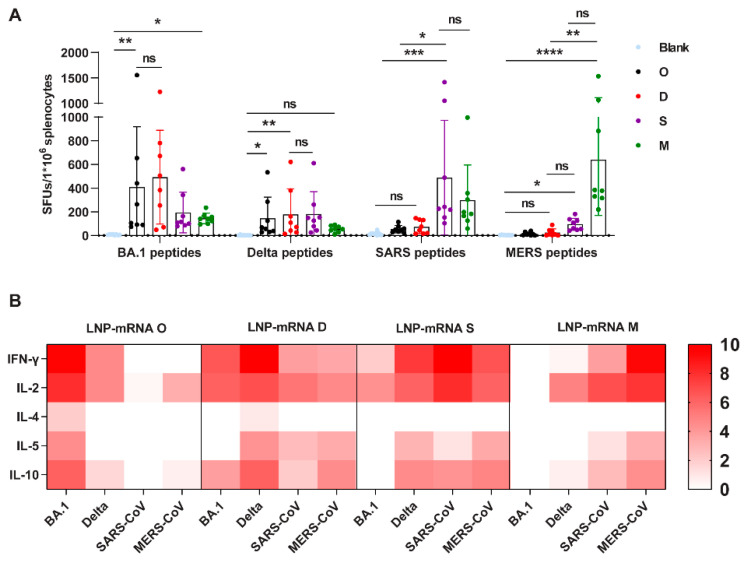
T-cell responses induced by the four monovalent mRNA vaccines. The number of IFN-γ-secreting spike protein-specific T cells was detected using an ELISPOT assay (**A**), and the concentrations of secreted IFN-γ, IL-2, IL-4, IL-5, and IL-10 were compared and represented by a heatmap. The concentration of each cytokine was transformed by log2 (**B**). Dunn’s multiple comparison test was used to compare the differences in (**A**). Bars represent the mean with SD, ns, *p* > 0.05; *, *p* < 0.05; **, *p* < 0.01; ***, *p* < 0.001; ****, *p* < 0.0001.

**Figure 4 viruses-16-00484-f004:**
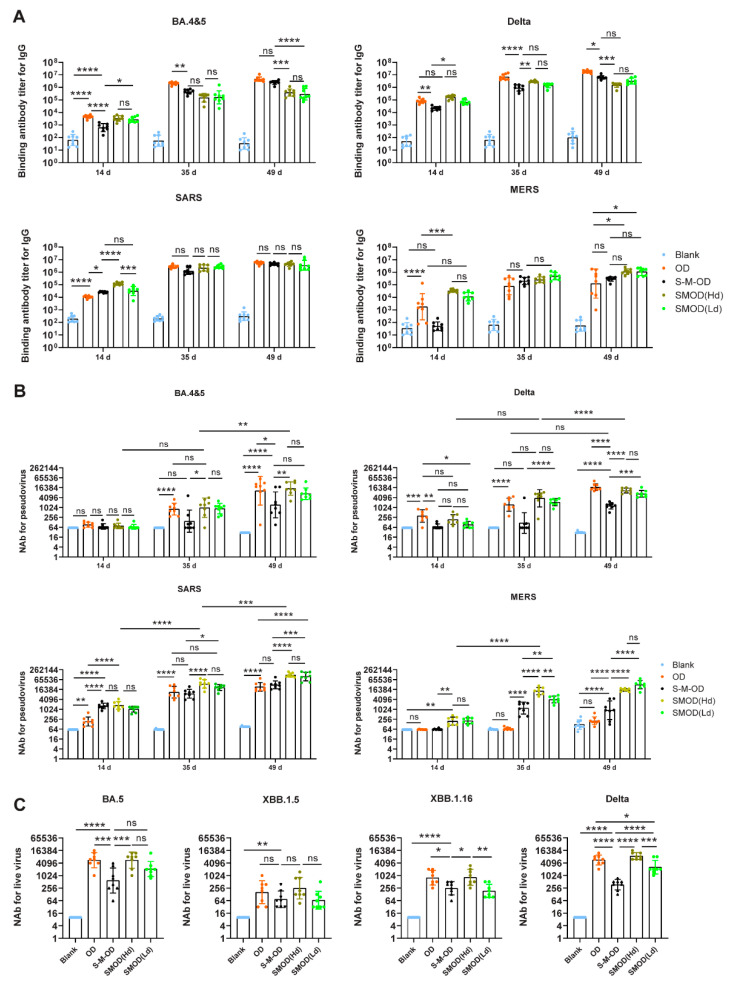
Antibody responses induced in mice immunized with sequential and cocktail immunization. Serum spike-specific IgG titers (**A**) and neutralizing (NtAb) titers against pseudoviruses (**B**) were measured on days 14, 35, and 49. NtAb titers against authentic viruses were measured on day 49 (**C**). *n* = 8 per group. Tukey’s multiple comparisons test was used to compare differences in (**A**–**C**). Bars represent the geometric mean with geometric SD, ns, *p* > 0.05; *, *p* < 0.05; **, *p* < 0.01; ***, *p* < 0.001; ****, *p* < 0.0001; Hd, high dose; Ld, low dose.

**Figure 5 viruses-16-00484-f005:**
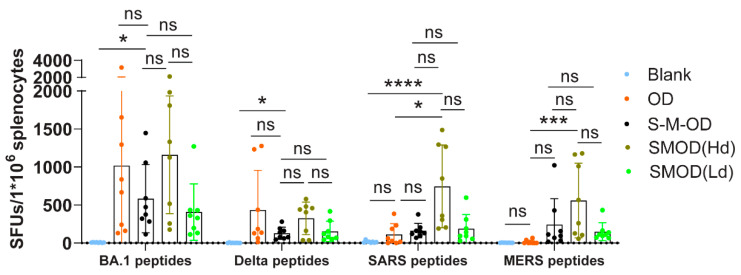
T-cell responses induced by mice immunized using sequential and cocktail immunization. The number of spike protein-specific T cells secreting IFN-γ was detected using an ELISPOT assay. Tukey’s multiple comparisons test was used to compare differences. Bars represent the mean with SD, ns, *p* > 0.05; *, *p* < 0.05; ***, *p* < 0.001; ****, *p* < 0.0001.

**Figure 6 viruses-16-00484-f006:**
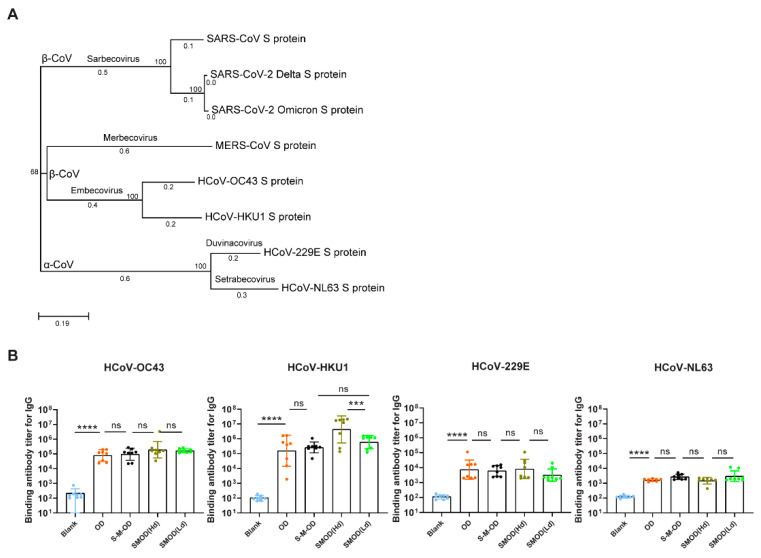
The highly pathogenic coronavirus vaccine induced binding antibody responses against other infectable human coronaviruses. (**A**) The genealogical tree was created using MEGA 7.0 software to sequence eight reference strains of human coronaviruses, including SARS-CoV-2 BA.4&5 variant (OM640071.14), Delta variant (OM858819.1), SARS-CoV (NC_004718.3), MERS-CoV (NC_019843.3), HCoV-OC43 (AY585228.1), HCoV-HKU1 (AY597011.2), HCoV-229E (AF304460), and HCoV-NL63 (AY567487.2), available in the NCBI database. (**B**) Detection of serum spike protein antigen-specific IgG against HCoV-OC43, HCoV-HKU1, HCoV-229E, and HCoV-NL63 using ELISA on day 49 after immunization. *n* = 8 per group. Tukey’s multiple comparisons test was adopted to compare the differences for (**B**). Bars represent the geometric mean with geometric SD, ns, *p* > 0.05; ***, *p* < 0.001; ****, *p* < 0.0001. IgG, immunoglobulin G; Hd, high dose; Ld, low dose.

## Data Availability

All data generated in this study are available from the corresponding author upon reasonable request.

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
