# Peer review of "A Cocktail of Lipid Nanoparticle-mRNA Vaccines Broaden Immune Responses against β-Coronaviruses in a Murine Model"

_viruses, 2024, doi:10.3390/v16030484_

Round 1

Reviewer 1 Report

Comments and Suggestions for Authors

In this manuscript, the authors describe results from experiments to assess the immunological response to LNP-mRNA vaccines against beta-coronaviruses, when delivered individually, sequentially, or as a cocktail in mice. The overall objectives were to support development of a pan beta-coronvarius vaccine to address emerging and possibly emerging human pandemic viruses.

The use of LNP-mRNA vaccines against coronaviruses in and of itself is not novel; however, there is clearly a need for improved vaccines and/or vaccine regimens to address emerging and re-emerging coronavirus strains. The data presented clearly demonstrates the ability of the individual vaccines to generate IgG, NtAb & cellular immune responses in mice to the corresponding virus and closely related viruses. The utilility of sequential or co-administration on broad spectrum immunological responses is also demonstrated. 

What's lacking from this analysis is the protective efficacy of the vaccines alone and in combination, duration of immunity, and any assessment of the potential for vaccine associated enhancement. The latter is of particular importance given the observation of cross-binding but non neutralizing antibodies and would be essential for preclinical efficacy assessment. Thus I would suggest that the authors include discussion on these topics which should also be addressed before moving to NHP studies.

The authors also must include more detailed information on the mouse studies and the statistical methods used in the materials and methods section. Details include source, strain, sex, and weight of animals; number of animals per group; vaccination route; vaccine formulations; vaccine administration; animal care and handling; and clinical observations. Clinical observation should also be included in the results and discussion.

Other items:

Line 50 - Coronaviruses bind (not binding).

Line 55 and 56 - sarbecoviruses and merbecoviruses were not previously defined. I suggest you include this in the initial introduction to the viruses.

Line 70 - delete the a from "and designing a shared antigenic....)

Line 90 - I'm not familiar with a eukaryocyte. Should be eukaryote? 

Line 91 - which contained not containing

Line 92 - O, D, S, and M should be defined here at first introduction.

Line 295 - delete f from fand

Line 297 - 17-fold not 17-old

Comments on the Quality of English Language

Aside from the minor gramatical errors noted above, the manuscript is perfectly comprehensible.

Author Response

Review 1

  1. What's lacking from this analysis is the protective efficacy of the vaccines alone and in combination, duration of immunity, and any assessment of the potential for vaccine cross-binding but non neutralizing antibodies and would be essential for preclinical efficacy assessment. Thus I would suggest that the authors include discussion on these topics which should also be addressed before moving to NHP studies.

Response:  We thank reviewer for this detailed advice. The discussion has been added in line 405-414, which were “In this study, S-M-OD and SMOD regimens both induced broad neutralizing antibodies against SARS, SARS-2 and MERS; amongst, NtAb titers in SMOD group are higher. Referring to current studies on the correlates of protection against coronaviruses, long-lasting and neutralizing antibody is the essential immune effector. “High binding and low neutralizing” might mediate antibody-dependent enhancement. Thus, we agree that SMOD regimen is more potential for immune protection against SARS, SARS-2 and MERS. However, due to limitations in experimental materials and methods, we were unable to detect the NtAb titers against other human coronaviruses. It should be addressed that further exploration is also needed to determine the protective efficacy in animal models.”

  1. The authors also must include more detailed information on the mouse studies and the statistical methods used in the materials and methods section. Details include source, strain, sex, and weight of animals; number of animals per group; vaccination route; vaccine formulations; vaccine administration; animal care and handling; and clinical observations. Clinical observation should also be included in the results and discussion.

Response:  We thank the reviewer for the advices. According to the reviewer’s suggestion, we have added the following content in the materials and methods section in the revised text.

1)    We had added the statistical methods as 2.7 in line 142-148 as

“ 2.7. Statistical Analyses 

All statistical analyses were conducted using GraphPad Prism 8 (GraphPad Soft-ware, Inc). Antibody titers were log-transformed and presented as the Geometric mean with geometric standard derivation (geometric SD). Turkey’s multiple comparisons test was used for the comparison of antibody responses in different groups. ELISPOT data were present as the mean with SD, and compared using Dunn’s multiple compar-ison test. P-value < 0.05 was considered statistically significant.”

2)     The detailed information of the mouse has been added in line 87-89   as “SPF grade BALB/c mice (6–8-week-old, female, 18-22g) were provided by and housed at the Laboratory Animal Resource Center, National Institute for Food and Drug Control were used as experimental animals.”

3)     The clinical observation has been added in line 180-181 as “During the immune procedure, all the mice grow well with no decrease in body weight”.

  1. Line 50 - Coronaviruses bind (not binding).

Response: We thank reviewer for this meaningful suggestion. We have revised the reference grammar in line 54.

  1. Line 55 and 56 - sarbecoviruses and merbecoviruses were not previously defined. I suggest you include this in the initial introduction to the viruses.

Response:We thank reviewer for this critical comment. We had added the definition of sarbecoviruses and merbecoviruses as “The Coronaviruses have caused several pandemics in 21th century and would continue to threat human health in future. Coronavirus is divided into four generas: α、β、γ、δ, and β-coronaviruses composes four lineages including A, B, C and D. The B lineage virus is also called sarbecovirus, including SARS-CoV, SARS-CoV-2 and its variants, and SARS‐related coronaviruses (SARSr‐CoVs) from bats and other animals. The C lineage viruses, also known as merbecovirus, include MERS-CoV and MERS-related coronaviruses (MERSr-CoVs) originated from bats” in line 38-44.

  1. Line 70 - delete the a from "and designing a shared antigenic....)

Response: We thank reviewer for this meaningful suggestion. We had not discussed clearly in the manuscript, and we had described it again as “for this reason, it is difficult to design a shared antigenic sequence to induce sufficient cross-protection against pathogenic merbecoviruses and sarbecoviruses” in line 74-76.

  1. Line 90 - I'm not familiar with a eukaryocyte. Should be eukaryote?

Response: We thank reviewer for this critical comment, we had changed “eukaryocyte” to “eukaryote” in line 95.

  1. Line 91 - which contained not containing

Response: This spelling mistake has been corrected as contained in line 96.

  1. Line 92 - O, D, S, and M should be defined here at first introduction.

Response: Thanks for your meaningful advice, we had defined the O,D,S,M as following: “ O (SARS-CoV-2 Omicron), D (SARS-CoV-2 Delta), S (SARS-CoV), or M (MERS-CoV)” in line 97-98.

  1. Line 295 - delete f from fand

Response: Thank you for your detailed comment. We had corrected the spelling error in line 312.

  1. Line 297 - 17-fold not 17-old

Response: We apologize for this mistake in this sentence. We have corrected the error in line 314. 

Reviewer 2 Report

Comments and Suggestions for Authors

In this study, four LNP-mRNA vaccines targeting the spike protein of SARS-CoV-2 Omicron, Delta, SARS-CoV, and MERS-CoV, were synthesized. All four LNP-mRNAs induced effective cellular and humoral immune responses against the corresponding spike protein antigens in mice. Moreover, the cocktail vaccination strategy induced more potent neutralizing antibodies and T cell responses against het erotypic viruses as well as broader antibody activity against pan-β-coronaviruses.

Several suggestions

1.      Line 93, please add [poly A adding] information since there is [a poly A tail] in line 140.

2.      Line 98, please describe more regarding the information of [liposomes] or add a reference.

3.      Line 99, please describe more regarding the information of [the average particle size of the LNP-mRNAs and evaluated their liposomal encapsulation rate] or add a reference.

4.      Line 157, please add a reference after [using a dsRNA residue detection kit], which is not mentioned in the [methods] section.

5.      Figure 2A, please mention the [dose] of the four LNP-mRNA vaccines, just like Table S2.

6.      Table S2, [[S, M, O, D 6 μg/LNP-mRNA] and [S, M, O, D 1.5 μg/LNP-mRNA]. 6 μg or 1.5 μg for each component or in total?

7.      Are there any side effects of these four LNP-mRNA vaccines when they were immunized in the mice?

8.      Figure 6, [binding antibody responses] may not be equal to neutralization antibodies. Neutralization tests using authentic viruses should not be possible for these viruses. Are there pseudoviral systems available for these viruses, just like Figures 2 and 4?

9.      Please discuss the possible explanations for the observation in lines 349-355.

10.  Typo errors: [105] in line 128; [CO2] in line 129.

Author Response

Review 2

  1. Line 93, please add [poly A adding] information since there is [a poly A tail] in line 140.

Response: We thank reviewer for this detailed advice. We had added [poly A adding] information as “(110 repeated A (adenine) added after 3'UTR)”in line 96.

  1. Line 98, please describe more regarding the information of [liposomes] or add a reference.

Response: We thank reviewer for this advice and had add the reference which is title “A Novel Amino Lipid Series for mRNA Delivery: Improved Endosomal Escape and Sustained Pharmacology and Safety in Non-human Primates” in line 105.

  1. Line 99, please describe more regarding the information of [the average particle size of the LNP-mRNAs and evaluated their liposomal encapsulation rate] or add a reference.

Response: We thank reviewer for this advice and had add the reference which title is “Are existing standard methods suitable for the evaluation of nanomedicines: some case studies” to determine the average particle size of the LNP-mRNAs in line 105. And also added the item number of the product to evaluated their liposomal encapsulation rate in line106.

  1. Line 157, please add a reference after [using a dsRNA residue detection kit], which is not mentioned in the [methods] section.

Response: We thank editor for this meaningful suggestion. We have added the detail information of [dsRNA residue detection kit] – “The dsRNA residues were measured by dsRNA residue detection kit (Vazyme, cat # DD3508-01)” in line 107-108.

  1. Figure 2A, please mention the [dose] of the four LNP-mRNA vaccines, just like Table S2.

Response: We thank editor for this comment. We have mentioned the dose of the four LNP-mRNA vaccines in line 202-203. And the sentence is “each mouse was injected intramuscularly in hind legs with 6μg LNP-mRNA/dose”.

  1. Table S2, [[S, M, O, D 6 μg/LNP-mRNA] and [S, M, O, D 1.5 μg/LNP-mRNA]. 6 μg or 1.5 μg for each component or in total?

Response: We thank reviewer for this advice and in Table S2, 6 μg and 1.5 μg all means for each component.

  1. Are there any side effects of these four LNP-mRNA vaccines when they were immunized in the mice?

Response: It’s a really good question. It’s very meaningful to assess the side effects mRNA vaccines as continuous severe side effects reported after mRNA vaccine administration in population. In mouse model, we found rare side effects after LNP-mRNA immunization. To be specific, all mice immunized were in good condition with healthy fur, diet, and weight. While it should be noted that we found only one mouse in high dose SMOD group with megalosplenia. We have added the clinical observation in line 180-181.

  1. Figure 6, [binding antibody responses] may not be equal to neutralization antibodies. Neutralization tests using authentic viruses should not be possible for these viruses. Are there pseudoviral systems available for these viruses, just like Figures 2 and 4?

Response: We thank reviewer for this critical comment. In this study, we have also attempted to test the pseudo-neutralizing antibodies against OC43, 229E, NL63 and HKU1. However, there is a huge difficulty to develop these methods, because the host receptors are unclear and the host cells need time to be well screened. We will continue to figure out these problems and strive to provide a detailed supplement to these results in the future.

  1. Please discuss the possible explanations for the observation in lines 349-355.

Response: We fully agree with reviewer. We had made detail discussion in the revised manuscript in line 367-376.

“Mice were immunized with LNP-mRNAs encoding spike from Omicron, Delta, SARS and MERS. The results showed that immunization with Omicron spike LNP-mRNA or Delta spike LNP-mRNA alone was difficult to induce a high NtAb response against both Omicron and Delta. This is consistent with the results of current reports that pre-Omicron variants, including WT, Alpha, Beta, Delta, possess certain degree of cross-neutralizing activity to each other, and real-world data also showed that the first-generation SARS-2 vaccines targeting ancestral virus was still able to provide effective protection against the Alpha and Beta variants; whereas the Omicron variant, gained more than 30 mutation sites, remains a poor cross-immune response to the pre-Omicron.”

  1. Typo errors: [105] in line 128; [CO2] in line 129.

Response: We thank editor for this comment. We have revised it accordingly in 137 and 138.